# Human SCARB2 Acts as a Cellular Associator for Helping Coxsackieviruses A10 Infection

**DOI:** 10.3390/v15040932

**Published:** 2023-04-08

**Authors:** Shu-Ling Yu, Nai-Hsiang Chung, Yu-Ching Lin, Yi-An Liao, Ying-Chin Chen, Yen-Hung Chow

**Affiliations:** 1Institute of Infectious Disease and Vaccinology, National Health Research Institutes, Zhunan Town, Miaoli County 350, Taiwan; julian0709@nhri.edu.tw (S.-L.Y.); cnc781005@nhri.edu.tw (N.-H.C.); yuch03@nhri.edu.tw (Y.-C.L.); onurluvly0504@gmail.com (Y.-A.L.); chin77@nhri.edu.tw (Y.-C.C.); 2Graduate School of Life Sciences, National Defense Medical Center, Taipei 114, Taiwan; 3Graduate Program of Biotechnology in Medicine, Institute of Molecular Medicine, National Tsing Hua University, Hsinchu 300, Taiwan; 4Institute of Molecular and Cellular Biology, National Tsing Hua University, Hsinchu 300, Taiwan; 5Graduate Institute of Biomedical Sciences, China Medical University, Taichung 404, Taiwan

**Keywords:** coxsackieviruses A10, human SCARB2, transgenic mice, HFMD

## Abstract

Coxsackievirus A10 (CVA10) causes hand, foot, and mouth disease (HFMD) and herpangina, which can result in severe neurological symptoms in children. CVA10 does not use the common enterovirus 71 (EV71) receptor, human SCARB2 (hSCARB2, scavenger receptor class B, member 2), for infection but instead uses another receptor, such as KREMEN1. Our research has shown that CVA10 can infect and replicate in mouse cells expressing human SCARB2 (3T3-SCARB2) but not in the parental NIH3T3 cells, which do not express hSCARB2 for CVA10 entry. Knocking down endogenous hSCARB2 and KREMEN1 with specific siRNAs inhibited CVA10 infection in human cells. Co-immunoprecipitation confirmed that VP1, a main capsid protein where virus receptors for attaching to the host cells, could physically interact with hSCARB2 and KREMEN1 during CVA10 infection. It is the efficient virus replication following virus attachment to its cellular receptor. It resulted in severe limb paralysis and a high mortality rate in 12-day-old transgenic mice challenged with CVA10 but not in wild-type mice of the same age. Massive amounts of CVA10 accumulated in the muscles, spinal cords, and brains of the transgenic mice. Formalin inactivated CVA10 vaccine-induced protective immunity against lethal CVA10 challenge and reduced the severity of disease and tissue viral loads. This is the first report to show that hSCARB2 serves as an associate to aid CVA10 infection. hSCARB2-transgenic mice could be useful in evaluating anti-CVA10 medications and studying the pathogenesis induced by CVA10.

## 1. Introduction

Coxsackievirus A10 (CVA10) belongs to the Picornavirus family, genus Enterovirus, and is a single-stranded RNA virus [1]. Over the past few decades, hand, foot, and mouth disease (HFMD) outbreaks associated with CVA10 have been reported in several countries, including France, Vietnam, and Uruguay [2,3,4]. CVA10-associated HFMD is especially severe in younger children [5,6]. Clinical manifestations have been reported, such as onychomadesis [7], herpangina, hyperkalemia, or death [7]. Although enterovirus 71 and Coxsackievirus A16 (CVA16) were previously believed to be the main causative agents of HFMD [8], the increased frequency of CVA10-associated HFMD outbreaks worldwide has challenged this view [5,9]. With no effective vaccine or antiviral agent, HFMD caused by CVA10 has become a significant public health concern worldwide.

Several animal models have been established to investigate the pathogenesis of CVA10 and to evaluate the efficacy of vaccines or antivirals, including mice [10,11,12,13,14], gerbils [15], and non-human primates. In suckling mice, challenge with a mouse-adapted strain of CVA10 shows mild pathological characteristics during the young-age period [11,16]. However, when a very high dose of the mouse-adapted strain of CVA10 (~5 × 10^9^ of 50% of cell culture infective dose, CCID50, per mouse) was injected into a local strain of mice during the newborn-age period, and a severe pathological response was observed [17]. These inadequate responses could not be correlated with the actual pathogenesis induced by CVA10 or the efficacy of antiviral medications. Therefore, establishing a relevant and useful animal model will help us further understand clinically manifested CVA10 infections.

KREMEN1 (KRM1) is the entry receptor for the largest receptor group of hand, foot-and-mouth-disease-causing viruses, which includes CVA2-6, A8, A10, and A12 [18,19]. KRM1 is a cell surface molecule that regulates WNT signaling by binding to DKK and LRP5/6, promoting the uptake of this complex through clathrin-mediated endocytosis [20]. The extracellular domain of KRM1 directly binds to CVA10 virions, facilitating virus entry into the cells [19].

Human scavenger receptor class B, member 2 (hSCARB2), has been identified as the cellular receptor for EV71 and CVA16 [21]. A type II glycoprotein of SCARB2 is expressed in many tissues, primarily in the limiting membranes of cell lysosomes and endosomes [22,23]. The expression of the hSCARB2 receptor in unsusceptible cell lines was found to facilitate infections by EV71 and CVA16 strains, resulting in the development of cytopathic effects [24]. We have created a transgenic mouse model expressing the hSCARB2 gene (hSCARB2-Tg) and reported severe hind limb paralysis, HFMD syndrome, and even death after EV71 or CVA16 infection during the young-age period [25]. hSCARB2-Tg mice serve as a model for studying EV71 and CVA16-mediated pathogenesis and evaluating anti-EV71 vaccines.

In this study, we discovered that CVA10 physically interacts with hSCARB2 and KRM1, which promotes viral infection in RD cells. CVA10 infection was inhibited by transfection with siRNAs that specifically target endogenous hSCARB2 and KRM1 expression. Infection of a clinically isolated strain of CVA10 in young-aged hSCARB2-Tg mice confirmed the occurrence of severe HFMD-associated pathology and even death. Finally, the experimental FI-CVA10 vaccine was shown to protect hSCARB2-Tg mice from the lethal challenge of CVA10. These results indicate that hSCARB2-Tg mice could be an ideal model for studying CVA10-induced pathogenesis and evaluating the efficacy of antiviral medications.

## 2. Materials and Methods

### 2.1. Ethics Statement

All the experiments were conducted according to the guidelines of the Laboratory Animal Center of the National Health Research Institutes (NHRI), Taiwan. Animal use protocols have been reviewed and approved by the NHRI Institutional Animal Care and Use Committee (approved protocol no. NHRI-IACUC-107153-M1-AE-S01. Date of approval: 6 February 2023).

### 2.2. Cells and Viruses

NIH3T3 mouse fibroblast cells (ATCC No. CRL-1658) were purchased from the American Type Culture Collection (ATCC, Manassas, Virginia, USA). 3T3-SCARB2 cells, which are derived from NIH3T3 transfected with a plasmid carrying human SCARB2 cDNA to express hSCARB2 (3T3-SCARB2), were generated in our previous work [26]. Medical Research Council cell strain-5 (MRC-5; ATCC No. CCL-171), a diploid cell line composed of fibroblasts, originally developed from ATCC, was purchased from the Bioresource Collection and Research Center (BCRC, BCRC No. 60023) of Taiwan. Human rhabdomyosarcoma (RD; BCRC No. 60113) cells derived from a human rhabdomyosarcoma were purchased from BCRC of Taiwan. These cell lines were cultured in a DMEM medium with 10% fetal bovine serum (Gibco, Thermo Fisher Scientific, Waltham, MA, USA). They were maintained in a 37 °C incubator equilibrated with 5% CO_2_. The CVA10_M2014 isolate was obtained from the National Cheng Kung University Hospital. CVA10 virus stocks were collected from the supernatants of infected RD or MRC-5 cells at 3 days post-infection (DPI). The virus stocks were stored at −80 °C. The supernatants containing virus particles were purified using 10–60% continuous sucrose gradient ultracentrifugation according to a protocol described previously [27]. The fractions were subjected to SDS-PAGE and then silver-stained. The titer of virus stocks was tested in RD cells using a standard plaque-forming assay [26]. The purified particles of CVA10 from infected MRC-5 cells were allowed to adsorb to aluminum phosphate (alum) at room temperature for 3 h before immunization. Moreover, 0.1 mL (1 × 10^7^ pfu + 30 μg alum) per dose of FI-CVA10 was prepared for immunization.

### 2.3. Antibodies and Western Blotting

EV71-specific mouse monoclonal antibody Mab979 was purchased from R&D system, Inc., Minneapolis, MN, USA. Anti-CVA10/CVA6 polyclonal rabbit antibody (GTX132346), which cross-reacts to the VP1 of CVA6 and CVA10; anti-CVA6 polyclonal rabbit antibody (GTX132348), which specifically reacts to the VP0 and VP2 of CVA6; and rabbit antibody against GAPDH were purchased from GeneTex, Inc., Irvine, CA, USA. Goat anti-hSCARB2 antibody was purchased from Invitrogen, Thermo Fisher Scientific, Waltham, MA, USA (PA5-19111). Antibodies specific to KRM1 were purchased from GeneTex, Inc. (GTX31876). The expression of viral capsid protein, hSCARB2, KRM1, and housekeeping protein GAPDH in CVA10-infected cells was assessed by Western blot. Cell lysates were prepared by treatment of 1–2 × 10^6^ cells in 100 μL of ice-cold lysis buffer (0.5% sodium deoxycholate, 0.1% sodium dodecyl sulfate (SDS), 0.5% NP-40, 50 mM TRIS, 150 mM NaCl) with the addition of a protease inhibitors cocktail (Roche, Basel, Switzerland) and 1 mM PMSF (Merck Sigma-Aldrich, Burlington, Massachusetts, USA). The lysates were centrifuged for 20 min at 10,000 rpm at 4 °C to sediment the cell debris. The cell lysate was subjected to SDS-polyacrylamide gel electrophoresis (SDS-PAGE) (Amersham Biosciences, Buckinghamshire, UK). This entailed loading 10 μg of the cell lysate mixed with loading dye per well of a 10% SDS-PAGE. Following electrophoresis at 100 V for 90 min in 1× Tris-glycine SDS-running buffer, the resolved proteins were transferred onto a nitrocellulose membrane (Hybond-ECL, Amersham Biosciences). The protein-containing membrane was soaked in 5% skim milk in PBS, pH 7.4, for 30 min at room temperature and then washed three times with 10 mL of assay buffer [PBS, pH 7.4 containing 0.05% Tween 20]. The membrane was incubated with 1:1000 diluted anti-CVA6/CVA10, 1:1000 diluted anti-hSCARB2, anti-KRM1, or 1:5000 diluted anti-GAPDH antibody, for 14–16 h at 4 °C and subsequently washed five times with 15 mL of 1× PBS plus 0.05% Tween 20 (PBS-T) followed by incubation with anti-rabbit secondary antibody conjugated with horse radish peroxidase (HRP) (1:10,000; GeneTex, Inc., Cat. GTX213110-01) or with anti-goat secondary antibody conjugated with HRP (1:5000; CHEMICON, Tokyo, Japan, Cat. #AP106P). After 1 h of incubation, the membrane was washed three times with PBS-T, the Millipore ECL substrate (Merck Millipore, Cat. WBKLS0500) was layered onto the membrane, and then the signal was detected using an Amersham Imager 600. When necessary, the membranes were stripped with restoring buffer (Pierce Biotechnology, Inc, Thermo Fisher Scientific) and used again with another antibody.

### 2.4. Co-Immunoprecipitation

Cell lysates (50 µg) from RD cells that had been infected with a multiplicity of infection (MOI) = 0.05 of CVA10 and cultured on 10 cm dishes for 24 h were mixed with 5 μL of protein G magnetic beads (Cytiva, Washington, DC, USA) and 1 µL of pull-down rabbit antibody specific to CVA6/CVA10, hSCARB2 (NOVUS Biologicals, Littleton, CO, USA), KRM1 (GeneTex, Inc.), or isotype antibody, which is specific to GAPDH (GeneTex, Inc.), respectively, at 4 °C for 6 h. After five washes with lysis buffer, lysate–magnetic beads were resuspended in 30 μL SDS loading buffer, boiled for 5 min, and then subjected to SDS-PAGE and Western blotting with primary antibody reacting to CVA/CVA10, hSCARB2, KRM1, or GAPDH. Detection with anti-hSCARB2 primary antibody (R&D, AF1966) used anti-goat IgG conjugated with HRP. The other detection was achieved using an appropriate anti-rabbit antibody conjugated with HRP (GeneTex, Inc., Rabbit EasyBlot kit, GTX225856-01). Cell lysates from 3T3-SCARB2 cells in the presence or absence of CVA10 infection were directly subjected to Western blotting, as described above. Each experiment has been repeated at least two times independently.

### 2.5. Reduction of Cellular Gene Expression by siRNA

RD cells at 80% confluency were transfected with oligonucleotides of siRNA specific to hSCARB2, KRM1 gene, or the negative control siRNA using the liposome–oligonucleotide transfection method. OptiMEM medium (Invitrogen) containing 100 picomoles of siRNA pre-mixed with 1 µL of Lipofectamine RNAiMAX Transfection Reagent (Invitrogen, Cat: 13-778-150) was added per well of a 12-well plate and incubated at 37 °C for 48 h in an incubator. After incubation, the cells were infected with CVA10 (MOI = 0.05) in serum-free culture medium followed by 1 h incubation at 37 °C before washing three times with a serum-free culture medium. After 24 h of incubation, the supernatant and cell lysates were prepared for a plaque-forming assay and the Western blot. Negative control siRNA, two hSCARB2 siRNA, s2651(S1) and s2652(S2), and two KRM1 siRNAs, s38393(K1) and s38394(K2) (Appendix A), were purchased from Ambion in life tech (Thermo Fisher Scientific).

### 2.6. CVA10 Infection in hSCARB2-Tg Mice

hSCARB2-Tg mice in the C57BL/6 background generated by our group were maintained by cross-mating hSCARB2-Tg individuals to obtain inbred mice [25]. Twelve-day-old Tg or wild-type (WT, C57BL/6) mice were inoculated subcutaneously (s.c.) with CVA10 or PBS alone. The mice were monitored daily for pathological signs and were sacrificed at various times post-inoculation to detect the viral amounts in the tissues. The severity of central nervous system (CNS) disease was scored from 0 to 5 using the following criteria [25]; 5 = severe front and rear limb paralysis (LP) and no movement, 4 = moderate 2 rear LP and hesitant movement, 3 = one rear LP with bending legs, 2 = mild rear limb bent, 1 = slightly rear limb bent, and 0 = normal movement. LP is defined as the rigidness of mouse legs that are hesitant to move. To evaluate the anti-CVA10 vaccine’s protective efficacy, 2-day-old hSCARB2-Tg mice were inoculated subcutaneously (s.c.) with 1 × 10^7^ pfu FI-CVA10 or an equal volume of PBS alone at day 2 and day 8 and then infected s.c. with 1.7 × 10^7^ pfu of CVA10 at day 12. The mice were monitored daily for disease severity, body weight, and survival rate for 15 days. The mice were sacrificed at various times post-inoculation, and their tissues were collected to detect the viral amounts using real-time RT-PCR and for immunohistochemistry staining.

### 2.7. Real-Time RT-PCR

The total RNA was purified from the tissues using TRIZOL reagent (Invitrogen) following the manufacturer’s instructions. The total RNA was reverse-transcribed into cDNA using random primers (Genomics BioSci&Tech, Taipei, Taiwan) with reverse transcriptase (Bionovas, Toronto, ON, Canada). The cDNA was subjected to quantitative PCR analysis (the LightCycler^®^ 480 SYBR Green Real-Time PCR system) using the primer pairs specific to the target genes. The CVA10-F and CVA10-R primer pair, probe of CVA10-P, mouse β-actin-F, and β-actin-R primer pair used as the internal control are listed in Table 1. The reagent for CVA10 quantitative PCR used KAPA PROBE FAST qPCR Master Mix (2×) (Kit Cat. KK4701), and β-actin detection used KAPA SYBR^®^ FAST qPCR Master Mix (2×) Kit (Cat. KK4609). The conditions used for CVA10 PCR were: 95 °C for 10 min, followed by 45 cycles of 95 °C for 3 s at 65 °C for 20 s. The conditions used for β-actin PCR were: 95 °C for 3 min, followed by 40 cycles of 95 °C for 10 s at 65 °C for 20 s and 72 °C for 2 s, followed by incubation at 72 °C for 2 min. The number of cycles required to amplify the target gene was obtained. The relative expression of the target gene was calculated as follows: the individual Ct obtained from the experimental group or control group was subtracted by its respective Ct (β-actin), and then 2^Normalized Ct (target gene from the sample without drug treatment)^ was divided by 2^Normaliszed Ct (target gene from the sample with drug treatment)^ [28]. All primer sets were synthesized commercially by Genomics BioSci&Tech.

### 2.8. ELISA

To detect CVA10-binding antibodies in sera, 96-well plates were coated with 100 μL per well of 1 × 10^5^ pfu of CVA10 in carbonate coating buffer. The serum samples collected from immune mice were inactivated at 56 °C for 30 min. Two-fold serial dilution of the sera was performed beginning from a 20-fold initial dilution. The diluted sera were added to the wells and incubated at room temperature for 2 h. After washing with PBS-T, HRP-conjugated donkey anti-mouse IgG antibodies were added to the wells for 45 min. The reaction was developed through incubation with 100 μL of TMB substrate (3, 3′, 5, 5′-etramethyllbenzidine) for 20 min in the dark and terminated by adding 50 μL of 2 N H_2_SO_4_. The optical density at 450 nm was determined using a microplate absorbance reader (SPECTRA, MAX2, M2). The end-point dilution, where the OD450 score was more than 2-fold of the OD450 of the background control wells, was determined as the titer of anti-CVA10 IgG.

### 2.9. Virus Neutralization Assay

Heat-inactivated serum samples were serially diluted two-fold with the fresh cell-culture medium. CVA10 virus (100 pfu) was added to each tube containing a 1:4 initiated dilution of serially diluted serum, 40 μL per tube, and incubated at 4 °C for 1 h. The virus–serum mixtures were added to 6-well plates seeded with 5 × 10^5^ per well of RD cells, and the plaque-forming assay was performed. The 50% neutralization inhibition dose (ID50) was calculated as the reciprocal of the serum dilution that yielded a 50% reduction in the plaque formation that referenced the plaque numbers generated in the tested wells infected with CVA10 pre-incubated with 1:4 diluted normal mouse serum.

### 2.10. Enzyme-Linked Immunosorbent Spot Assay

Tg or WT (C57BL/6) mice (6 to 8 weeks old) were inoculated s.c. with two doses of 5 μg/mouse of FI-CVA10 vaccine or PBS with a 2-week interval. The mice were then sacrificed at 2 weeks after the second immunization. RBC-free splenocyte suspensions (5 × 10^5^) prepared from individual mice were seeded in individual wells of 96-well filtration plates (Merck Millipore) precoated with capturing monoclonal antibodies for murine IL-4 or IFN-γ (0.5 μg/well) (Cat. No. 16-7041-68 or 16-7313-68, respectively, eBioscience, Thermo Fisher Scientific) and blocked with conditioned medium (LCM) for 1 h at room temperature. The splenocytes were added to 10^6^ pfu per well of UV-inactivated CVA10 dissolved in LCM (100 μL). The splenocytes added with Con A (10 μg/mL) were used as a positive control. Unstimulated splenocytes were used as a negative control. The plates were maintained in a 37 °C incubator equilibrated with 5% CO_2_ for 48 h. The individual wells of the ELISPOT plates were washed three times with PBS-T, and then 0.2 µg of the corresponding biotinylated detection monoclonal antibodies specific for IL-4 and IFN-γ were added to detect the respective cytokines. After 2 h of incubation at room temperature, the plates were washed, and 100 µL of streptavidin–alkaline phosphatase (1:250 dilution) was added to the individual wells. The plates were incubated at room temperature for 45 min. Finally, the plates were washed four times with the wash buffer, and 100 µL of AEC (3-amine-9-ethylcarbazole, Sigma-Aldrich) substrate was added to each well and allowed to react for 30 min at room temperature in the dark. The plates were then washed with water and air-dried overnight, and the spots on each well were scored using an immunospot counting reader (Immunospot, Cellular Technology Ltd.). The results were expressed as the number of cytokine-secreting cells per 5 × 10^5^ splenocytes seeded in the initial culture.

### 2.11. Statistical Analysis

A two-way ANOVA test was used to analyze the results shown in Figure 1B. A one-way ANOVA test was used to analyze the results from Figures 3B–D, 4A–C, 5A–C, 6B–D and 7A–C. A paired Student’s *t*-test was used to analyze the results shown in Figure 7D. The results were considered statistically significant when *p* < 0.05. The symbols *, **, *** and **** are used to indicate *p*  < 0.05, *p*  <  0.01, *p*  < 0.001, and *p*  < 0.0001, respectively.

## 3. Results

### 3.1. hSCARB2 Expression in 3T3-SCARB2 Facilitates CVA10 Infection

Our previous study identified hSCARB2 as a receptor for EV71 and demonstrated its ability to facilitate infection in mouse cell lines expressing hSCARB2 (3T3-SCARB2) [27]. To investigate whether hSCARB2 could also support CVA10 infection, RD, 3T3-SCARB2, and parental NIH3T3 cells were inoculated with the CVA10 virus, and viral replication was examined. Cytopathic effects were observed in 3T3-SCARB2 and RD cells after 3 days of inoculation. Cultured supernatants and lysates from CVA10-infected cells were collected and subjected to measurement of de novo produced CVA10 by plaque-forming assay. CVA10 was produced in the lysate and supernatant of 3T3-SCARB2 cells but not in NIH3T3 cells (Figure 1A).

The amounts of produced CVA10 were significantly higher in both supernatant and cell lysate of 3T3-SCARB2 cells at day 3 post-viral inoculation, compared to no virus was detected in both supernatant and cell lysate collected from NIH3T3 cells (Figure 1B). The amount of CVA10 production in the supernatant and cellular lysate of RD cells used as a control was also calculated (Figure 1C). CVA6/CVA10-specific polyclonal antibody, which could detect the VP1 of CVA6 and CVA10 in the lysates derived from CVA6 and CVA10-infected RD cells, respectively, but not EV71 in the lysate derived from EV71-infected RD cells (Appendix A), was used to detect CVA10 VP1 in the lysate of infected cells. We observed that 3T3-SCARB2 cells infected with CVA10 could express VP1 with a molecular weight of 36 KDa at 24 h post-infection (h.p.i) and gradually increased at 48 and 72 h.p.i. In contrast, no VP1 was detected at any time points in NIH3T3 cells infected with CVA10 (Figure 1D). These findings were corroborated by either plaque formation or the expression of viral capsid proteins, which were seen in 3T3-SCARB2 but not in NIH3T3 cells, confirming that mouse cells were unsusceptible to CVA10 infection and that hSCARB2 expression had converted them into a susceptible cell line.

### 3.2. Interaction of CVA10 with hSCARB2 and KRM1

To examine whether hSCARB2 interacted with CVA10 during infection, the co-immunoprecipitation of hSCARB2 using a specific antibody following the detection of CVA10 and KRM1 in CVA10-infected or uninfected RD cells was performed. After 24 h of infection, cell lysates were mixed with magnetic bead-conjugated anti-hSCARB2 or KRM1-specific antibodies for immunoprecipitation. Blotting with the CVA6/CVA10 antibody showed that VP1 was detected in the hSCARB2-pulled-down precipitates from CVA10-infected RD cells but not in the precipitates from uninfected RD cells. The direct detection of 36 KDa of CVA10 VP1 from the lysates derived from CVA10-infected RD cells is shown in the middle panel of Figure 2A. Because KRM1 had been identified as a cellular receptor of CVA10 [19], we further examined whether KRM1 was present in the hSCARB2-pulled-down precipitates, and it was found that KRM1 could not be detected in the co-immunoprecipitates prepared from CVA10-infected or uninfected RD cells. The detection of 48 KDa of KRM1 from the lysates derived from CVA10-infected and uninfected RD cells was seen (upper panel of Figure 2a). An 82 KDa band of hSCARB2 was detected in all of the hSCARB2-pulled-down precipitates and lysates (lower panel of Figure 2a). In the KRM1-pulled-down precipitates, VP1 was co-precipitated in the precipitates derived from CVA10-infected lysate, whereas no detection of VP1 was observed in the precipitated derived from the uninfected lysate (middle panels of Figure 2b).

Interestingly, hSCARB2 was weakly detected in the KRM1-pulled-down precipitates from the CVA10-infected lysates but not from the uninfected lysates. The detection of hSCARB2 from the lysates derived from CVA10-infected and uninfected RD cells was observed (upper panels of Figure 2b). As expected, KRM1 was detected in all of the KRM1-pulled-down precipitates and lysates (lower panel of Figure 2b). However, due to the nonspecific binding activity of the polyclonal CVA6/CVA10 antibody, the binding of KRM1 and hSCARB2 in the lysate during the preparation of co-precipitates was competed. The other possibility was that the interaction of KRM1 with hSCARB2 was steric-dependent. The confirmation of binding KRM1 in the hSCARB2-pulled-down precipitates was changed during the process in the denature condition of electrophoresis, and subsequently, the blotting antibody was not able to recognize it. Thus, we were unable to detect KRM1 in the hSCARB2-pulled-down precipitates from the CVA10-infected lysate (upper panel of Figure 2a). The isotypic antibody (specific to GAPDH) pulled-down precipitates from the lysate derived from 8 and 24 h of CVA10-infected or uninfected RD cells were also prepared and applied to immunoblot. However, hSCARB2, KRM1, or CVA10 VP1 could not be detected (upper, middle, or lower panels of Figure 2c). The direct detection of hSCARB2 and KRM1 in the lysates from all infected and uninfected RD cells was shown. VP1 was observed in the CVA10-infected lysate but not in the uninfected lysate (Figure 2c).

Additionally, GAPDH was observed in all the GAPDH-pulled-down precipitates and lysates (lower panel of Figure 2c). These results indicate that CVA10 VP1 can bind to both hSCARB2 and KRM1 during infection. Interestingly, KRM1 can physically associate with hSCARB2 in the presence of CVA10 infection in RD cells.

### 3.3. Inhibition of CVA10 Infection by Knocking down hSCARB2 Expression

To confirm that hSCARB2 can support CVA10 infection, the researchers knocked down endogenous hSCARB2 expression in RD cells using specific siRNAs (S1 and S2, Appendix A) before infecting the cells with CVA10. The results showed that CVA10 VP1 expression was significantly reduced by S2 and mildly reduced by S1 compared to negative siRNA-transfected cells (first panel of Figure 3A). The expression of cellular hSCARB2 was also significantly reduced by S1 and S2 compared to negative siRNA-transfected cells (third panel of Figure 3A), indicating that hSCARB2 siRNA impaired CVA10 infection in RD cells. The researchers also knocked down endogenous KRM1 expression in RD cells using specific siRNAs (K1 and K2, Appendix A) after infecting the cells with CVA10. Results showed that CVA10 VP1 expression was almost completely reduced by K1 and K2, respectively, compared to negative siRNA-transfected cells, indicating that CVA10 infection was blocked (first panel of Figure 3A). However, the level of cellular KRM1 was not significantly reduced by K1 and K2 (second panel of Figure 3A).

In parallel, the levels of de novo synthesized CVA10 in the supernatants and the cellular lysates of infected cells were also examined. The results revealed that the levels of de novo synthesized CVA10 in the supernatants and cellular lysates were significantly reduced upon transfection with S1 and S2 siRNA, compared to the levels of CVA10 obtained from the supernatants and the cellular lysates of the cells treated with negative siRNA (Figure 3B,C, respectively). The total levels of de novo synthesized CVA10 calculated by summing the levels obtained from the supernatants and cellular fractions showed that CVA10 production was inhibited by S1 and S2 siRNA treatment (Figure 3D). Similarly, the levels of de novo synthesized CVA10 in the supernatants and the cellular fractions were significantly reduced (Figure 3B,C, respectively), and the inhibition of total CVA10 levels by individual K1 and K2 siRNA was observed (Figure 3D).

Interestingly, we observed that the reduction level of synthesized CVA10 obtained from the cells transfected with mixed siRNA of S1/K1 or S2/K2 was similar to the reduction level of synthesized CVA10 obtained from the cells transfected with a single siRNA of K1 or K2. A previous study showed that Rac1-dependent endocytosis and Rab5-dependent intracellular trafficking are required by both EVA71 and CVA10 [29]. Rac1 is a GTPase that participates in macropinocytosis or other pathways of endocytosis by regulating cytoskeletal organization and kinase activity [30]. KRM1 internalized via clathrin-mediated endocytosis has been reported [29]. hSCARB2 associated with clathrin-mediated endocytosis in EV71 infection has also been reported [26]. These results suggest that the cellular hSCARB2 and KRM1 are interactively regulated. hSCARB2 may associate with KRM1 following the triggering of the entry process during CVA10 infection.

### 3.4. CVA10 Induces a Severe Neurological Disease in hSCARB2-Tg Mice

We previously created a lineage of hSCARB2-Tg mice and used them to study EV71-mediated pathogenesis [25,31] and to test the efficacy of anti-EV71 medications [32,33]. This study investigated whether hSCARB2-Tg mice could serve as a reliable model for CVA10 infection. We subcutaneously inoculated mice of different ages and genotypes with varying amounts of CVA10 and monitored them daily for pathological responses. We found that the severity of neurological symptoms, such as limb paralysis and body weight change, was age- and viral-amount-dependent. The disease score was higher in older newborn Tg and WT mice, and a higher viral load was required to induce severe disease resulting in higher mortality. We also observed that newborn hSCARB2-Tg mice were more sensitive to CVA10 than the same-age WT mice (Appendix A).

In a study of 12-day-old mice, the highest disease score was observed in hSCARB2-Tg mice challenged with 1.72 × 10^7^ pfu CVA10, with a middle score in hSCARB2-Tg mice challenged with 10-fold less CVA10, and the lowest score in WT mice receiving 1.72 × 107 pfu CVA10 (Figure 4a). Similarly, the most severe weight loss was observed in hSCARB2-Tg mice receiving 1.72 × 10^7^ pfu, with a middle severity in hSCARB2-Tg mice receiving 1.72 × 10^6^ pfu, and the lowest severity in WT mice receiving 1.72 × 10^7^ pfu CVA10 (Figure 4b). The hSCARB2-Tg mice infected with 1.72 × 10^6^ pfu or 1.72 × 10^7^ pfu CVA10 by subcutaneous injection had higher susceptibility leading to death (25% and 0% survival rate, respectively) at 9 days post-infection (DPI), compared to WT mice challenged with the same doses, which showed 90% and 100% survival rates, respectively (Figure 4c). However, the older hSCARB2-Tg mice (2-week-old and up) infected with the same doses of CVA10 showed very mild or no symptoms and eventually recovered.

To examine the distribution of CVA10 in the tissues of 12-day-old hSCARB2-Tg and WT mice on day 3 post-infection, a real-time RT-PCR of the CVA10 VP1 mRNA region was performed. Tissues from WT mice of the same age without viral infection were included as a background control. CVA10 VP1 signals were detected in the skeletal muscle, brain, and spinal cord of infected Tg mice, with a significantly higher viral load in the muscle compared to WT mice (Figure 5a). Moreover, infected Tg mice had a higher viral load in the brain than WT mice (Figure 5b). However, the levels of virions in the spinal cord were similar between Tg and WT mice (Figure 5c). These findings indicate that the accumulation of CVA10 in the CNS and muscle tissues may contribute to the disease severity observed in infected Tg mice (Figure 4).

### 3.5. Immunization of Anti-CVA10 Vaccine Protects Mice from Live CVA10 Challenge

The effectiveness of anti-CVA10 medications was further evaluated using hSCARB2-Tg mice. A flowchart of the immunization protocol, which involved administering a two-dose formalin-inactivated CVA10 (FI-CVA10) vaccine followed by a live CVA10 challenge, was shown (Figure 6a). The results showed that 100% of FI-CVA10-immunized Tg mice survived, while all PBS-treated Tg mice died (Figure 6b). Mild changes in body weight and very low disease scores, with eventual recovery on day 6 post-infection, were observed in FI-CVA10-immunized Tg mice, in contrast to the highest body weight changes and most severe disease observed in PBS-treated Tg mice (Figure 6c,d, respectively). The amounts of CVA10 RNA in different tissues were examined on day 3 post-infection using quantitative RT-PCR. Compared to the tissue from the Tg mice that did not receive FI-CVA10 treatment, a reduction in accumulated CVA10 was observed in the muscle tissue of the Tg mice that received the FI-CVA10 vaccine, although this difference was not statistically significant (Figure 7a). The amounts of CVA10 RNA in the brain and spinal cord of Tg mice that were pre-immunized with FI-CVA10 were significantly reduced (Figure 7b,c, respectively).

The binding activity of FI-CVA10-immunized serum against CVA10 virions was observed (Appendix A). The neutralizing antibody titer was significantly raised (>128) in the serum of mice immunized with FI-CVA10 compared to the serum of non-immunized mice (Figure 7d). Additionally, induction of splenocytic IFN-γ and IL-4 secretion was observed in the FI-CVA10-immunized mice (Appendix A). These results support the induction of protective immunity by FI-CVA10 in newborn hSCARB2-Tg mice. Moreover, our findings demonstrate that hSCARB2-Tg mice serve as a reliable model to evaluate the efficacy of CVA10 vaccines, as the prototypic FI-CVA10 vaccine demonstrated effectiveness in preventing CVA10 infection.

## 4. Discussion

A previous study indicated that hSCARB2 is the cellular receptor for EV71 [24] but not for CVA10 infection [18]. However, our study showed that 3T3-SCARB2 cells, but not parental NIH3T3 cells, were susceptible to CVA10 infection (Figure 1a and b). Although the yield of CVA10 production in 3T3-SCARB2 cells was 10,000 times lower than in RD cells (Figure 1c), it prompted us to investigate the role of hSCARB2 in CVA10 infection. Specific siRNA targeting hSCARB2 could not completely inhibit CVA10 infection in RD cells, even though the expression level of SCARB2 was impaired (Figure 3). This indicates that hSCARB2 is not a major receptor but rather serves as an associate factor that supports CVA10 infection through other functional receptors. We also examined KRM1, the reported major receptor in CVA10 infection, and found that inhibiting endogenous KRM1 expression by specific siRNA was able to inhibit CVA10 entry into RD cells. However, the treatment of both siRNAs targeting hSCARB2 and KRM1 did not show an additive or synergistic inhibition in CVA10 infection (Figure 3), indicating that the pathway of KRM1-mediated CVA10 entry is bound with hSCARB2. To ensure the role of SCARB2 as an associate factor for CVA10, a co-immunoprecipitation study confirmed that the interaction of hSCARB2 and KRM1 occurred in the presence of CVA10 (Figure 2). Further studies are needed to elucidate the mechanism of hSCARB2 and KRM1 and CVA10 VP1 interaction and determine whether other cellular factors are involved in this interaction.

This report presents the first evidence of hSCARB2 involvement in CVA10 infection. Previous studies have confirmed that hSCARB2-Tg animals are an appropriate experimental model for EV71 and CVA16 infections [25]. Mouse models, including inbred BALB/c, ICR, a local strain of mice, and outbred gerbils, ranging from one to fourteen days old, have been reported for mouse-adapted CVA10 infection [12]. We demonstrated that hSCARB2-Tg mice at 12 days old are more susceptible to CVA10 infection than the reported newborn mouse models, leading to severe neurological pathology and death (Figure 5). CVA10 infection in hSCARB2-Tg mice exhibits bi-pathological tropism in the CNS and peripheral sites, including muscle, similar to human infections, whereas it also causes severe damage to tissues such as herpangina, onychomadesis, and CNS complications [34,35]. The viral load in the CNS and peripheral tissues of hSCARB2-Tg mice may correspond to virulence and lead to lethal neurological diseases. Other factors, such as the age of the host, which exhibits a different gene expression profile in cells, may also affect susceptibility. However, the level of receptor expression is unlikely to be the cause, as the expression of hSCARB2 was similar in newborns and adult Tg mice [25].

Preventive vaccines are the most effective and economical means of controlling EV-related diseases. Although an EV71-inactivated vaccine is currently available on the market, it does not confer cross-protection against other EV serotypes, such as CVA10. Numerous inactivated CVA10 vaccines, including formalin-inactivated vaccines (clinical isolate TA151R [10], clinical isolate CVA10-FJ-01 [11]), and β-propiolactone-inactivated vaccines (original strain CVA10/Kowalik, clinical isolate CVA10/S0148b), have been evaluated for their ability to produce high titer antibodies that can protect newborn suckling mice against viral challenge [16]. Immunizing hSCARB2-Tg mice with FI-CVA10 followed by direct challenge with live CVA10 induced strong immunoprotection, as evidenced by the high titer of neutralizing antibody produced (Figure 7d), reduced disease severity, and complete prevention of animal death (Figure 6c–d and 6b, respectively). The FI-CVA10-immunized spleen also demonstrated the induction of CVA10-specific cellular immunity, as judged by lymphocytic IFN-γ and IL-4 secretions (Appendix A).

In summary, we have confirmed the role of hSCARB2 as an associate factor for supporting CVA10 infection, in which KRM1 physically interacts with hSCARB2 during viral entry. Our cell culture results pave the way to develop an animal model based on hSCARB2-Tg mice for CVA10 infection. hSCARB2-Tg mice elicit high susceptibility for severe CNS symptoms, leading to death after infection. This breakthrough may overcome the limitation of current animal model applications, prolong the time frame of mice age for disease induction, and induce host-adapted immunity by vaccination. It prompts us to plan further prophylactic and therapeutic approaches to control coxsackievirus infection.

## Figures and Tables

**Figure 1 viruses-15-00932-f001:**
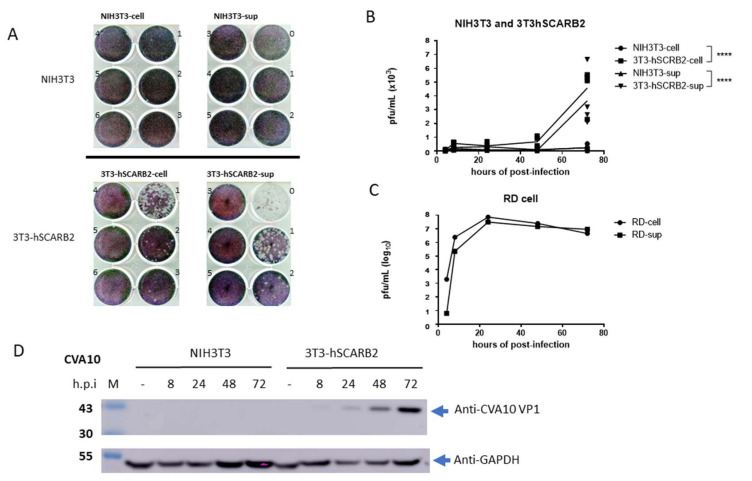
Replication of CVA10 in hSCARB2-expressing cells. Mouse NIH3T3 and 3T3-SCARB2 and human cell line, RD, were infected with 0.1 MOI of CVA10 and then cultured for up to 72 h. The cultured supernatants (sup) and cell lysates (cell) were collected at 8, 24, 48, and 72 h, respectively, and subjected to the plaque-forming assay in RD cells described in the Materials and Methods. (**A**) Numbers 0, 1, 2, and up to 6 marked on each well were represented as 10^0^, 10^1^, 10^2^, and up to 10^6^, respectively, of serial dilution fold of the collected samples that were subjected to the assay. The photography of forming plaques was taken at 72 h after incubation. (**B**) The time course of produced CVA10 from the supernatants and lysate parts was calculated and shown. (**C**) The time course of CVA10 production in the supernatants and lysate parts of RD cell was also calculated. (**D**) Immunoblots of CVA10 VP1 and GAPDH in the lysates were collected at 8, 24, 48, and 72 h.p.i. by anti-CVA6/CVA10 and GAPDH antibodies, respectively. The symbol **** was used to indicate *p*  < 0.0001, respectively.

**Figure 2 viruses-15-00932-f002:**
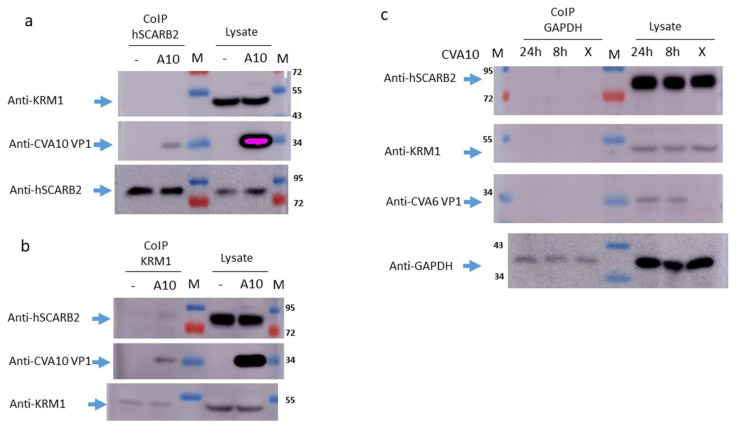
Association of hSCARB2 with KRM1 in CVA10-infected RD cells. RD cells were inoculated with MOI = 0.1 of CVA10 and cultured for 8 and 24 h before lysate preparation. Uninfected RD cells were cultured in parallel. Next, 50 μg cell lysates were prepared and mixed with the magnetic bead-conjugated antibody against (**a**) hSCARB2, (**b**) KRM1, and (**c**) isotypic antibody specific to GAPDH, individually, for immunoprecipitation (CoIP). The co-precipitates were subjected to Western blot against associated hSCARB2, CVA10 VP1, and KRM1 proteins. Lysates (20 μg) from infected and uninfected RD cells were directly subjected to immunoblot to detect the investigated proteins.

**Figure 3 viruses-15-00932-f003:**
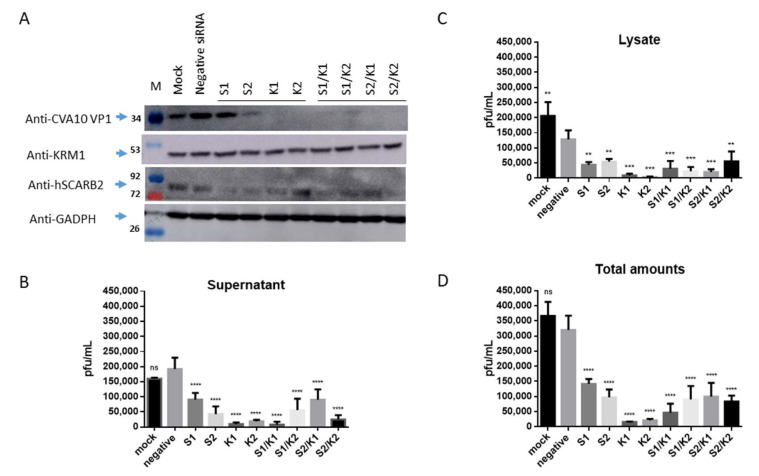
The reduction of endogenous hSCARB2 and KRM1 expression impacted CVA10 infection. RD Cells were transfected individually with 50 pmoles of siRNA specific to hSCARB2 (s2651(S1) and s2652(S2)), KRM1 (s38393(K1) and s38394(K2)), or mixed specific siRNA with equal amounts (50 + 50 pmoles) of S1 + K1, S1 + K2, S2 + K1, S2 + K2, or negative siRNA, followed by 48 h of incubation. Infection of siRNA-treated cells with CVA10 (MOI = 0.05) and then cultured for another 24 h then proceeded. (**a**) Western blotting with the respective antibody examined the expression level of hSCARB2 or KRN1 in the cells. (**b**) The supernatant and (**c**) lysate were collected and subjected to a plaque-forming assay to detect the amounts of produced CVA10, and the results were shown. (**d**) Total amounts of CVA10 production using the sum of the viral amounts from (**b**,**c**) are shown. The symbols **, *** and **** were used to indicate *p*  <  0.01, *p*  < 0.001, and *p*  < 0.0001, respectively.

**Figure 4 viruses-15-00932-f004:**
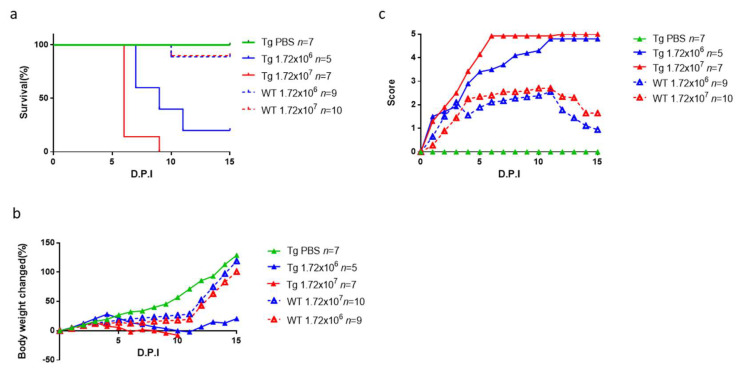
Pathogenesis and lethality of hSCARB2-Tg mice infected with CVA10. Scoring of (**a**) disease score, (**b**) loss of body weight, and (**c**) survival rate in 12-day-old hSCARB2-Tg and WT mice injected s.c. with 1.72 × 10^6^ and 1.72 × 10^7^ pfu of CVA10, individually, were monitored daily (day post-infection, D.P.I.) for 15 days and assessed following the criteria described in the Materials and Methods section. The same age of Tg mice injected s.c. with PBS as control was included. The number (*n*) of mice per group was shown.

**Figure 5 viruses-15-00932-f005:**
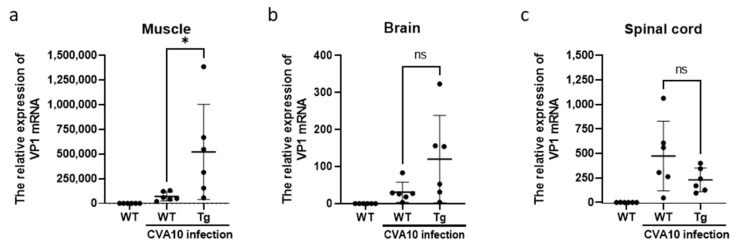
Viral distribution in the tissues and organs of CVA10-infected mice. Twelve-day-old WT and hSCARB2-Tg mice were injected with 1.72 × 10^7^ pfu of CVA10 s.c. On day 3, post-viral infection, RNA extracted from the (**a**) muscle, (**b**) brain, and (**c**) spinal cord of the mice were subjected to quantitative RT-PCR analysis using primers specific to the VP1 region. β-actin gene expression in each tissue was used as the internal control. A schematic representation of the VP1 gene expression and the statistical average from six mice per group, as described in the Materials and Methods section, is shown. The symbols * and ns were used to indicate *p*  < 0.05 and no significant difference, respectively.

**Figure 6 viruses-15-00932-f006:**
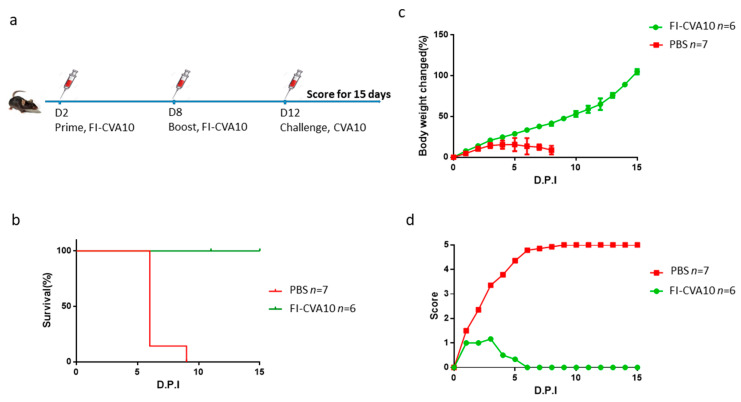
Protection of hSCARB2-Tg mice from CVA10 challenge by FI-CVA10 vaccine. (**a**) The flowchart of vaccine immunization following the challenge of CVA10 in Tg mice was shown. One-day-old Tg mice were s.c. immunized with two doses of FI-CVA10 vaccine or PBS on day 2 and day 8, following s.c. injection of 1.72 × 10^7^ pfu of CVA10 on day 11 was performed. Daily recording of (**b**) percentage of mice survival, (**c**) loss of body weight, and (**d**) disease score for 15 days of hSCARB2-Tg mice receiving PBS or FI-CVA10 was performed. Data represent the mean scores obtained from the individual number of (*n*) mice per group.

**Figure 7 viruses-15-00932-f007:**
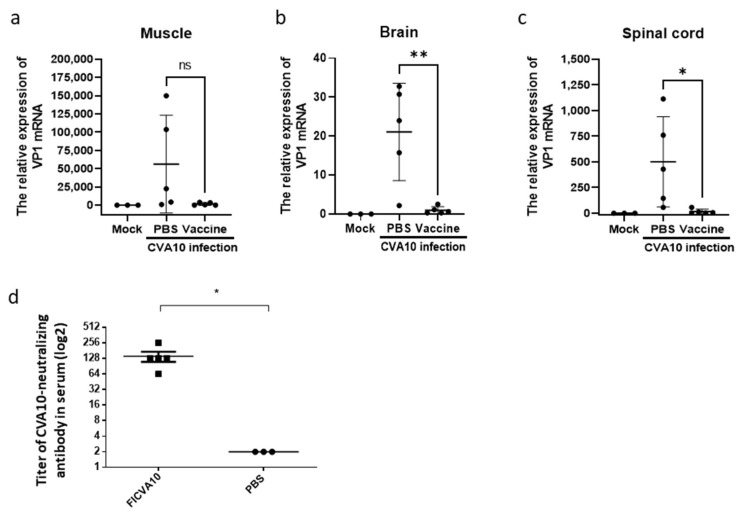
Reduction in viruses in the tissues of CVA10-infected mice pre-receiving FI-CVA10 vaccine. On day 3 post-viral infection, RNA was extracted from the (**a**) muscle, (**b**) brain, and (**c**) spinal cord of Tg mice, which had been pre-immunized with PBS or two-dose of FI-CVA10 (described in the legend of Figure 6), and were subjected to quantitative RT-PCR analysis using primers specific to the VP1 region. RNA from the same age of Tg mice without any treatment (Mock) as background control was included. β-actin gene expression in each tissue was used as the internal control to calculate the relative expression of VP1 mRNA. A schematic representation of the VP1 gene expression and the statistical average from five mice per group from the FI-CVA10-treated group and three mice per group from the PBS-treated group (minimizing the number of animals used in the test) is shown. (**d**) Tg mice (6–8 weeks old) pre-immunized with PBS or two-dose of FI-CVA10 at a 2-week interval. Serum samples collected from individual Tg mice on day 28 post the second shot of FI-CVA10 were assayed for the titer of anti-CVA10 neutralizing antibodies. The results were expressed as titers for each test sample. Bars correspond to the mean titers for each experimental group. The symbols *, **, and ns were used to indicate *p*  < 0.05, *p*  < 0.01, and no significant difference, respectively.

**Table 1 viruses-15-00932-t001:** List of the sequence of primer pairs specific to target genes.

Primers	Target Gene	Sequences (5′->3′) of siRNAs
CVA10-F	CVA10 VP1	CGGTCCCTTTCATGTCACCA
CVA10-R		ACTCTCACTGCAAAGGTGCC
Probe of CVA10-P	CVA10 VP1	HEX-AAACTCACTGACCCTCCTGCACAAGTCTCA-BHQ1
β-actin-F	Mouse β-actin	ACCAACTGGGACGACATGGAGAAA
β-actin-R		TAGCACAGCCTGGATAGCAACGTA

## Data Availability

Not applicable.

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
