# Peer review of "Human SCARB2 Acts as a Cellular Associator for Helping Coxsackieviruses A10 Infection"

_viruses, 2023, doi:10.3390/v15040932_

Round 1

Reviewer 1 Report

Coxsackievirus A10 (CVA10) has been shown to use kringle containing transmembrane protein 1 (KREMEN1, KRM1) as an uncoating receptor.  The authors of this paper have demonstrated that the expression of human scavenger receptor B2 (hSCARB2) in mouse cells and in mice allows CVA10 to replicate and knockdown of hCARB2 in human cells inhibited replication. 

There is evidence that CVA10 VP1 interacts with hSCARB2 and with KREMEN1 using immunoprecipitation from infected human cells.  The assay was to immunoprecipitate hSCARB2 or KRM1 or GAPDH from CVA10 infected RD cell lysates with a magnetic bead conjugated antibody and then western blot to detect hSCARB2, KRM1 and CVA10 capsid protein.  The experiment was able to detect CVA10 immunoprecipitation with both hSCARB2 and KRM1 immunoprecipitation.  The hSCARB2 immunoprecipitation did not detect KRM1 but the KRM1 immunoprecipitation did have detectable hSCARB2.  The authors discuss this in the results section 359-363 “due to the non-359 specific binding activity of the polyclonal CVA6/CVA10 antibody, the binding of KRM1 and hSCARB2 in the lysate during the preparation of co-precipitates was competed,”.  I’m not sure what that means and I think this is an important finding as the results with the hSCARB2 transgenic mice and mouse cells indicate that at some stage in replication, hSCARB2 is playing an important role. If the conditions used compete with the ability to detect KRM1 in the hSCARB2 immunoprecipitate, why is it possible to detect KRM1 in the KRM1 immunoprecipitate. The CVA10 capsid is well detected in both of the western blots.  Indeed, I would suggest that a better control than GAPDH would be to immunoprecipitate the CVA10 capsid and screen for hSCARB2 and KRM1. 

Knockdown of hSCARB2 and KRM1 with two siRNAs each had the puzzling result that as expected, knockdown of siRNA greatly reduced the pfu/well of RD cells but, unexpectedly, did not affect the level of KRM1 present in the cells.  On the other hand, knockdown of hSCARB2 was effective at reducing the amount of cellular SCARB2 and reduced the level of CVA10 replication but not as much as KRM1 knockdown.  The affect of dual SCARB2 and KRM1 knockdown was not additive but seemed in two combinations to increase the level of CVA10 replication beyond what was seen with the KRM1 knockdowns alone. Co-precipitation of hSCARB2 in CVA10 infected cells by KRIM1, suggests that CVA10 may bind both. This is suggestive but it seems very odd that it is not seen in co-immunoprecipitation by hSCARB2.  This should be discussed in section 3.3, lines 399-403.  Is it possible that the functions in the cell of these two proteins are regulating processes important for successful replication of CVA10?  It is a very different process to add a protein (hSCARB2) to a murine cell rather than to eliminate a protein (KRM1 or hSCARB2) with a real function from human cells.  In the murine cells, the addition of hSCARB2 may not greatly affect normal cellular processes. I agree that this indicates hSCARB2 has a function in CVA10 replication regardless of an interaction with KRM1.

Inoculation of young mice with CVA10 produced much more severe disease and high mortality if the mice were transgenic for hSCARB2.  These effects in mice transgenic for hSCARB2 could be greatly reduced by vaccination prior to infection. This is a very telling observation, leading to a much improved murine model for this and perhaps other CVA viruses. 

There is no institutional review statement for the use of vertebrate animals in research.

Minor points:
On line 51-52, on the discovery of KRM1 as a receptor for CVA2, 6, 8, 10 and 12, reference 20 giving the original discovery should be included with reference 18 which covers the structure. 

On lines 91-92 and figure 1 legend, what cell line was used for the plaque forming assay.  It is not given in reference 27.

In figure 1, I know that the time course of infection in RD cells (Fig. 1c) is a demonstration of CVA10 replication in human cells but the units of the y axis in both Fig. 1b and c should be the same.  They are pfu/ml in Fig.1b and pfu/well in Fig. 1c.  This is necessary to confirm the increase in titer with expression of hSCARB2 is significant.

The method used for calculating the relative expression of target genes on lines 196-200 should be referenced: Livak KJ, Schmittgen TD. Analysis of relative gene expression data using real-time quantitative PCR and the 2(-Delta Delta C(T)) Method. Methods. 2001 Dec;25(4):402-8. doi: 10.1006/meth.2001.1262. PMID: 11846609.

On line 325, SCARB2 should be hSCARB2.

On lines 336-337, there is a partial sentence that is fairly cryptic.

Author Response

Coxsackievirus A10 (CVA10) has been shown to use kringle containing transmembrane protein 1 (KREMEN1, KRM1) as an uncoating receptor.  The authors of this paper have demonstrated that the expression of human scavenger receptor B2 (hSCARB2) in mouse cells and in mice allows CVA10 to replicate and knockdown of hCARB2 in human cells inhibited replication. 

There is evidence that CVA10 VP1 interacts with hSCARB2 and with KREMEN1 using immunoprecipitation from infected human cells.  The assay was to immunoprecipitate hSCARB2 or KRM1 or GAPDH from CVA10 infected RD cell lysates with a magnetic bead conjugated antibody and then western blot to detect hSCARB2, KRM1 and CVA10 capsid protein.  The experiment was able to detect CVA10 immunoprecipitation with both hSCARB2 and KRM1 immunoprecipitation.  The hSCARB2 immunoprecipitation did not detect KRM1 but the KRM1 immunoprecipitation did have detectable hSCARB2.  The authors discuss this in the results section 359-363 “due to the non-359 specific binding activity of the polyclonal CVA6/CVA10 antibody, the binding of KRM1 and hSCARB2 in the lysate during the preparation of co-precipitates was competed,”.  I’m not sure what that means and I think this is an important finding as the results with the hSCARB2 transgenic mice and mouse cells indicate that at some stage in replication, hSCARB2 is playing an important role.

Reply: We had discussed vigorously in the text regarding that the hSCARB2 immunoprecipitation did not detect KRM1 but the KRM1 immunoprecipitation did have detectable hSCARB2. The description was written as “due to the nonspecific binding activity of the polyclonal CVA6/CVA10 antibody, the binding of KRM1 and hSCARB2 in the lysate during the preparation of co-precipitates was competed. The other possibility was that the interaction of KRM1 with hSCARB2 was steric-dependent. The confirmation of binding KRM1 in the hSCARB2-pulled-down precipitates was changed during the process in the denature condition of electrophoresis, and subsequently the blotting antibody was not able to recognize it. Thus, we were unable….” in the line 23, section 3.2 of the result.   

 If the conditions used compete with the ability to detect KRM1 in the hSCARB2 immunoprecipitate, why is it possible to detect KRM1 in the KRM1 immunoprecipitate. The CVA10 capsid is well detected in both of the western blots.  Indeed, I would suggest that a better control than GAPDH would be to immunoprecipitate the CVA10 capsid and screen for hSCARB2 and KRM1. 

Reply: Thanks for your advice. Indeed, we had performed CVA10 VP1-pulled-down precipitates using the polyclonal CVA6/CVA10 antibody and following detected both hSCARB2 and KRM1 in the precipitates. The result of blotting was failed due to the nonspecific binding activity elicited in the polyclonal CVA6/CVA10 antibody as mentioned in the text.

Knockdown of hSCARB2 and KRM1 with two siRNAs each had the puzzling result that as expected, knockdown of siRNA greatly reduced the pfu/well of RD cells but, unexpectedly, did not affect the level of KRM1 present in the cells.  On the other hand, knockdown of hSCARB2 was effective at reducing the amount of cellular SCARB2 and reduced the level of CVA10 replication but not as much as KRM1 knockdown.  The affect of dual SCARB2 and KRM1 knockdown was not additive but seemed in two combinations to increase the level of CVA10 replication beyond what was seen with the KRM1 knockdowns alone. Co-precipitation of hSCARB2 in CVA10 infected cells by KRIM1, suggests that CVA10 may bind both. This is suggestive but it seems very odd that it is not seen in co-immunoprecipitation by hSCARB2.  This should be discussed in section 3.3, lines 399-403.  Is it possible that the functions in the cell of these two proteins are regulating processes important for successful replication of CVA10?  It is a very different process to add a protein (hSCARB2) to a murine cell rather than to eliminate a protein (KRM1 or hSCARB2) with a real function from human cells.  In the murine cells, the addition of hSCARB2 may not greatly affect normal cellular processes. I agree that this indicates hSCARB2 has a function in CVA10 replication regardless of an interaction with KRM1.

Inoculation of young mice with CVA10 produced much more severe disease and high mortality if the mice were transgenic for hSCARB2.  These effects in mice transgenic for hSCARB2 could be greatly reduced by vaccination prior to infection. This is a very telling observation, leading to a much improved murine model for this and perhaps other CVA viruses. 

  • Reply: Thanks for your suggestion. We had discussed vigorously in section 3.3, the last paragraph as” Previous study showed that Rac1-dependent endocytosis and Rab5-dependent intracellular trafficking are required by both EVA71 and CVA10 [29]. Rac1 is a GTPase, participated in macropinocytosis or other pathways of endocytosis, through regulating cytoskeletal organization and kinase activity [30]. KRM1 was internalized via clathrin-mediated endocytosis was reported [29]. hSCARB2 associated with clathrin-mediated endocytosis in EV71 infection was also reported [26], These results suggest that the cellular hSCARB2 and KRM1 are interactively regulated. hSCARB2 may associate with KRM1 following the triggering of the entry process during CVA10 infection.”

Minor points:
On line 51-52, on the discovery of KRM1 as a receptor for CVA2, 6, 8, 10 and 12, reference 20 giving the original discovery should be included with reference 18 which covers the structure. 

Reply: We had edited it as suggestion.

On lines 91-92 and figure 1 legend, what cell line was used for the plaque forming assay.  It is not given in reference 27.

Reply: We had described it in the legend of Fig. 1 and in the section 2.2 of Materials and Methods.

In figure 1, I know that the time course of infection in RD cells (Fig. 1c) is a demonstration of CVA10 replication in human cells but the units of the y axis in both Fig. 1b and c should be the same.  They are pfu/ml in Fig.1b and pfu/well in Fig. 1c.  This is necessary to confirm the increase in titer with expression of hSCARB2 is significant.

Reply: We had replotted the panel b and c to allow the y axis to be consistent. The unit as PFU/mL on the y axis was used in both panel b and c.

The method used for calculating the relative expression of target genes on lines 196-200 should be referenced: Livak KJ, Schmittgen TD. Analysis of relative gene expression data using real-time quantitative PCR and the 2(-Delta Delta C(T)) Method. Methods. 2001 Dec;25(4):402-8. doi: 10.1006/meth.2001.1262. PMID: 11846609.

 Reply: We had included the reference: Livak KJ, Schmittgen TD. Methods. 2001 Dec;25(4):402-8. doi: 10.1006/meth.2001.1262. PMID: 11846609, in the section 2.7 of Materials and Methods.

On line 325, SCARB2 should be hSCARB2.

 Reply: We had edited it as suggestion.

On lines 336-337, there is a partial sentence that is fairly cryptic.

 Reply: We had completed the sentence in the paragraph as “Because KRM1 had been identified as a cellular receptor of CVA10 [19], we further examined whether KRM1 was present in the hSCARB2-pulled-down precipitates, and it was found that KRM1 could not be detected in the co-immunoprecipitates prepared from CVA10-infected or uninfected RD cells.” in the line 9, section 3,2.

Reviewer 2 Report

Enteroviruses (EVs) are important human pathogens and deepening the knowledge of their cell receptors would promote the development of a model infection in small and easy-to-handle laboratory animals, which in turn would contribute to studying the pathogenesis of infection and the development of selective antiviral agents and effective vaccines.

There are several EVs that are capable of causing hand, foot, and mouth disease. They can be divided into 4 groups depending on their cellular receptor usage. Some use SCARB2 (for example EV-A71 and CVA16), others use KREMEN1 like CVA10, still others use CAR (some coxsackie B viruses) and still others use DAF/FcRn.

The authors of the submitted manuscript experimentally prove that CVA10 can infect and efficiently replicate in cells that express SCARB2, thus presenting the first report showing the role of SCARB2 as an associate factor for efficient CVA10 replication. The authors also demonstrate that transgenic mice expressing human SCARB2 could be a useful in vivo model for studying CVA10 pathogenesis and for the development of vaccines and antivirals.

To my opinion, the presented research work is relevant and very significant. The manuscript presents a lot of experimental work allowing reliable results and firm conclusions. 

Nevertheless, I have some remarks and comments. Here they follow:

Major Remarks:

1. The Abstract: The abstract is a bit unclear to a general scientific reader who could be outside the topic for EV receptors in particular. I would suggest writing over the abstract and making it easier for reading and understanding without losing the scientific virtue of the work.

Line 18: Please clarify that parental NIN3T3 cells do not express these receptors for virus entry.

Line 20: Clarify that VP1 is the main capsid protein where virus receptors for attaching to the host cells are situated.

Line 20-21: Please specify that it is not the interaction, itself, between the virus and its cell receptor that results in limb paralysis. It is the efficient virus replication following virus attachment to its cellular receptor.

Line 21: Please specify the age of the transgenic mice. What does the term ‘young’ mice mean? Please specify that both transgenic and wild-type mice were inoculated on day 12 after birth. Please clarify to the general reader that at that age wild type mice are already insusceptible to coxsackievirus infection.

2. Materials and Methods. Cells and Viruses.

NIH3T3 cells and 3T3-SCARB2 cells should also be described.

Line 166: The age of mice at the time of inoculation should be indicated here.   

3. Results. Knocking down hSCARB2 inhibits CVA10 infection.

Using the third person singular to describe one's own results is not appropriate.

Line 433: RT-PCR does not count the number of virions.

4. Figures. Please mind the place of the figures in the manuscript.

Minor remarks:

1. The Abstract:

Line 15: enterovirus 17. It should be 71.

Line 16: …uses other receptors. I would recommend ‘….uses another receptor such as …..’.

Line 16, line 18: KRMEN1. I suppose it should be KREMEN1.

Line 17, line 18: an abbreviation is already introduced (on line 15) for human SCARB2, i.e. hSCARB2. Please use it all down the text.

Line 19: siRNA. I would suggest using the plural, i.e. siRNAs.

Line 23: vaccine-induced. I suppose there is no need for a hyphen.

2. The Introduction:

            Line 39: antiviral reagent. I would propose using antiviral agent’.

Line 41, line 55: CV-A10. Please use uniform nomenclature throughout the whole manuscript, i.e. CVA10 (without hyphenation).

Line 44: showed. Use the Present Simple Tense of the verb.

Line 50: clinically isolated. Do you mean clinically ‘manifested’?

Line 56: I would suggest starting a new paragraph for hSCARB2.

Line 57: a cellular receptor. I would suggest using the definite article, i.e. the cellular receptor

Line 60: coxsackievirus A16: the abbreviation CVA16 is already introduced (on line 31). Please use it all down the text.

Line 62: transgenic mouse. Please insert ‘model’ or use the plural, as you haven’t developed only one transgenic mouse.

Line 41: formalin-inactivated CVA10 vaccine. Please insert here the abbreviation FI-CVA10, as it is its first appearance in the text. It is used unexplained further in the text (e.g. on line 95).

3. Materials and Methods. Cells and Viruses. Please remove the hyperlinks in the paragraph.

            Line 84: There is no need RD cell to be in red.

Line 97: Supplementary Fig. 1A: There is no reason the figure to be introduced in Materials and Methods, as it describes results.

Line 138: Define the abbreviation for the multiplicity of infection before introducing it.

Line 166, line 168: Please define the abbreviations Tg, WT, s.c., CNS.

Line 185: Wouldn’t it be more readable if sequences were presented in a table?

4. Results.

            Line 312, line 313: Instead of 'parts' it would be more appropriate to use 'both supernatant and cell lysate'

Line 312-314: Written in this way, the sentence could be easily misunderstood and the reader could get a notion that a virus was harvested in NIH3T3 cell, too which wouldn’t be true. In fact, there is no virus and this is indicated on lines 310-311.

Line 315: I would suggest using cellular ‘lysates’ instead of ‘parts’.

Line 325: insert ‘h’ in front of SCARB2, i.e. hSCARB2.

Line 336-337: The last sentence in the paragraph is incomplete.

Line 389, line 390, line 394: use ‘lysates’ instead of fractions.

Line 417, line 420, line 423, etc.: Do not mingle between both terms, WT, and non-Tg mice. Use consistently only one of them throughout the text.

Line 439-440, line 445: use only the abbreviation.

Lines 444-447: The sentences almost repeat the sentences above.

5. Discussion.

            Line 485: No need for the Genitive here

Line 494: herpangina, onychomadesis – These could not be considered as severe damage to vital organs. Neither the oral mucosa, nor the nails can be considered as vital organs.

Line 520: I consider the term ‘non-EV71 coxsackievirus’ infection inappropriate.

6. Figures.

           Fig. 1: Expalian that ‘sup’ stands for supernatant. Write NIH3T3 instead of 3T3. Explain that numbers 0, 1, 2, etc. in Fig. 1a stand for virus dilution. Fig. 1b and 1c – please apply one and the same scale for ‘b’ and ‘c’, either PFU/mL or PFU/well. The caption of the figure, line 105 – What does (A) stand for? Line 110 (b) – define in the caption that this part of the figure concerns NIH3T3 and 3T3-SCARB2 cells.

Fig. 2: Explain the abbreviation CoIP. Use KREMEN1 or KRM1 throughout the whole of the text, but choose only one of them and do not minge between them.

            Fig. 4, caption, line 267: please use either WT mice, or non-Tg mice uniformly throughout the text but do not mingle between both terms.

Author Response

Major Remarks:

  1. The Abstract: The abstract is a bit unclear to a general scientific reader who could be outside the topic for EV receptors in particular. I would suggest writing over the abstract and making it easier for reading and understanding without losing the scientific virtue of the work.

Line 18: Please clarify that parental NIN3T3 cells do not express these receptors for virus entry.

Reply: Thanks for your suggestion. We had edited the sentence in the line 5 of abstract section as: “but not in the parental NIH3T3 cells which do not express hSCARB2 for CVA10 entry”.

Line 20: Clarify that VP1 is the main capsid protein where virus receptors for attaching to the host cells are situated.

Reply: We had edited the sentence in the line 8 of abstract section as: “confirmed that VP1, a main capsid protein where virus receptors for attaching to the host cells,”.

Line 20-21: Please specify that it is not the interaction, itself, between the virus and its cell receptor that results in limb paralysis. It is the efficient virus replication following virus attachment to its cellular receptor.

Reply: We had edited the sentence in the line 10 of abstract section as: “It is the efficient virus replication following virus attachment to it’s cellular receptor.”.

Line 21: Please specify the age of the transgenic mice. What does the term ‘young’ mice mean? Please specify that both transgenic and wild-type mice were inoculated on day 12 after birth. Please clarify to the general reader that at that age wild type mice are already insusceptible to coxsackievirus infection.

Reply: We had edited the sentence in the line 11-12 of abstract section as: “a high mortality rate in 12-day-old transgenic mice challenged with CVA10 but not in the same age of wild-type mice”.

  1. Materials and Methods. Cells and Viruses.

NIH3T3 cells and 3T3-SCARB2 cells should also be described.

Reply: We had described NIH3T3 cells and 3T3-SCARB2 cells in the Materials and Methods, Cells and Viruses section

Line 166: The age of mice at the time of inoculation should be indicated here.   

Reply: We had described in the line 1-3 of Materials and Methods, CVA10 infection in hSCARB2-Tg mice section.

  1. Results. Knocking down hSCARB2 inhibits CVA10 infection.

Using the third person singular to describe one's own results is not appropriate.

Reply: We had edited the sentence as “Inhibition of CVA10 infection by knocking down hSCARB2 expression

Line 433: RT-PCR does not count the number of virions.

Reply: We had edited the sentence in the line 9 and13, Immunization of anti-CVA10 vaccine protects mice from live CVA10 challenge of result section as “The amounts of CVA10 RNA”

  1. Figures. Please mind the place of the figures in the manuscript.

 Reply: We had reorganized the place of the figures in the manuscript.

Minor remarks:

  1. The Abstract:

Line 15: enterovirus 17. It should be 71.

Reply: We had corrected it.

Line 16: …uses other receptors. I would recommend ‘….uses another receptor such as …..’.

Reply: We had corrected it as suggestion.

Line 16, line 18: KRMEN1. I suppose it should be KREMEN1.

Reply: We had corrected it.

Line 17, line 18: an abbreviation is already introduced (on line 15) for human SCARB2, i.e. hSCARB2. Please use it all down the text.

Reply: We had corrected it as suggestion.

Line 19: siRNA. I would suggest using the plural, i.e. siRNAs.

Reply: We had corrected it as suggestion.

Line 23: vaccine-induced. I suppose there is no need for a hyphen.

Reply: We had corrected it.

  1. The Introduction:

            Line 39: antiviral reagent. I would propose using antiviral agent’.

Reply: We had corrected it as suggestion.

Line 41, line 55: CV-A10. Please use uniform nomenclature throughout the whole manuscript, i.e. CVA10 (without hyphenation).

Reply: We had corrected it in the whole manuscript.

Line 44: showed. Use the Present Simple Tense of the verb.

Reply: We had corrected it as suggestion.

Line 50: clinically isolated. Do you mean clinically ‘manifested’?

Reply: We had corrected it as suggestion.

Line 56: I would suggest starting a new paragraph for hSCARB2.

Reply: We had corrected it as suggestion.

Line 57: a cellular receptor. I would suggest using the definite article, i.e. the cellular receptor

Reply: We had corrected it as suggestion.

Line 60: coxsackievirus A16: the abbreviation CVA16 is already introduced (on line 31). Please use it all down the text.

Reply: We had corrected it as suggestion.

Line 62: transgenic mouse. Please insert ‘model’ or use the plural, as you haven’t developed only one transgenic mouse.

Reply: We had corrected it as suggestion.

Line 41: formalin-inactivated CVA10 vaccine. Please insert here the abbreviation FI-CVA10, as it is its first appearance in the text. It is used unexplained further in the text (e.g. on line 95).

Reply: We had corrected it as suggestion.

  1. Materials and Methods. Cells and Viruses. Please remove the hyperlinks in the paragraph.

Reply: We had corrected it as suggestion.

            Line 84: There is no need RD cell to be in red.

Reply: We had corrected it as suggestion.

Line 97: Supplementary Fig. 1A: There is no reason the figure to be introduced in Materials and Methods, as it describes results.

Reply: We had removed them as suggestion.

Line 138: Define the abbreviation for the multiplicity of infection before introducing it.

Reply: We had edited it in the line 1 of section 2.4. Co-Immunoprecipitation as: “multiplicity of infection (MOI) “

Line 166, line 168: Please define the abbreviations Tg, WT, s.c., CNS.

Reply: We had defined the abbreviations Tg, WT, s.c., CNS in the first description in the manuscript

Line 185: Wouldn’t it be more readable if sequences were presented in a table?

Reply: We had edited it and presented in the Table 1 which shown in the manuscript

  1. Results.

            Line 312, line 313: Instead of 'parts' it would be more appropriate to use 'both supernatant and cell lysate'

Reply: We had edited it as 'both supernatant and cell lysate' in the manuscript

Line 312-314: Written in this way, the sentence could be easily misunderstood and the reader could get a notion that a virus was harvested in NIH3T3 cell, too which wouldn’t be trueIn fact, there is no virus and this is indicated on lines 310-311.

Reply: We had edited it in the line of result section, hSCARB2 expression in 3T3-SCARB2 facilitates CVA10 infection, as 'compared to no virus was detected in both supernatant and cell lysate collected from NIH3T3 cells (Fig. 1b).'

Line 315: I would suggest using cellular ‘lysates’ instead of ‘parts’.

Reply: We had edited them as suggestion.

Line 325: insert ‘h’ in front of SCARB2, i.e. hSCARB2.

Reply: We had corrected it as suggestion.

Line 336-337: The last sentence in the paragraph is incomplete.

Reply: We had completed the sentence in the paragraph as “Because KRM1 had been identified as a cellular receptor of CVA10 [19], we further examined whether KRM1 was present in the hSCARB2-pulled-down precipitates, and it was found that KRM1 could not be detected in the co-immunoprecipitates prepared from CVA10-infected or uninfected RD cells.” in the line 9, section 3,2

Line 389, line 390, line 394: use ‘lysates’ instead of fractions.

Reply: We had edited them as suggestion.

Line 417, line 420, line 423, etc.: Do not mingle between both terms, WT, and non-Tg mice. Use consistently only one of them throughout the text.

Reply: We had corrected them throughout the text.

Line 439-440, line 445: use only the abbreviation.

Reply: We had edited them throughout the text.

Lines 444-447: The sentences almost repeat the sentences above.

Reply: We had deleted the repeated parts already. 

  1. Discussion.

            Line 485: No need for the Genitive here

Reply: We had edited it as suggestion.

Line 494: herpangina, onychomadesis – These could not be considered as severe damage to vital organs. Neither the oral mucosa, nor the nails can be considered as vital organs.

Reply: We had edited the sentence in the discussion section as” whereas it also causes severe damage to tissues such as herpangina, onychomadesis, and CNS complications [34, 35].”

Line 520: I consider the term ‘non-EV71 coxsackievirus’ infection inappropriate.

Reply: We had deleted the term ‘non-EV71” in the sentence  

  1. Figures.

           Fig. 1: Expalian that ‘sup’ stands for supernatant. Write NIH3T3 instead of 3T3. Explain that numbers 0, 1, 2, etc. in Fig. 1a stand for virus dilution.

Reply: We had described ‘sup’ stands for supernatant in the legend. The numbers 0, 1, 2, etc. in Fig. 1a stand for virus dilution had been described in the legend as” (a) Numbers 0, 1, 2, and up to 6 marked on each well were represented as 100, 101, 102, and up to 106, respectively, of serial dilution fold of the collected samples that were subjected to the assay.”

Fig. 1b and 1c – please apply one and the same scale for ‘b’ and ‘c’, either PFU/mL or PFU/well.

Reply: We had replotted the panel b and c to allow the y axis to be consistent. The unit as PFU/mL on the y axis was used in both panel b and c.

The caption of the figure, line 105 – What does (A) stand for? Line 110 (b) – define in the caption that this part of the figure concerns NIH3T3 and 3T3-SCARB2 cells.

Reply:  Because of typing error, we had deleted “(A)” in the line 105 of legend. We had edited the caption already to be consistent in the figure concerns NIH3T3 and 3T3-SCARB2 cells.

Fig. 2: Explain the abbreviation CoIP. Use KREMEN1 or KRM1 throughout the whole of the text, but choose only one of them and do not minge between them.

Reply: We had described the mean of abbreviation CoIP in the legend. We had edited it as KRM1 in the figure and in the text.

            Fig. 4, caption, line 267: please use either WT mice, or non-Tg mice uniformly throughout the text but do not mingle between both terms.

Reply: We had edited as “WT” mice throughout the text.
